# Male Partners’ Knowledge, Attitudes, and Perception of Women’s Breast Cancer in Abha, Southwestern Saudi Arabia

**DOI:** 10.3390/ijerph16173089

**Published:** 2019-08-25

**Authors:** Hassan M. Al-Musa, Nabil J. Awadalla, Ahmed A. Mahfouz

**Affiliations:** 1Department of Family and Community Medicine, College of Medicine, King Khalid University, Abha 61421, Saudi Arabia; 2Department of Community Medicine, College of Medicine Mansoura University, Mansoura 35516, Egypt; 3Department of Epidemiology, High Institute of Public Health, Alexandria University, Alexandria 21511, Egypt

**Keywords:** breast cancer, male knowledge, Saudi Arabia

## Abstract

*Background*: Breast cancer (BC) is ranked as the most frequently diagnosed cancer site among women in Saudi Arabia. Several studies in Saudi Arabia have reported low awareness of BC and significant obstacles to early presentation among Saudi women. A key sociocultural obstacle against breast screening and early detection of BC in several conservative cultures is that men manage women’s choices and activities. The aim of this research is to find out the key background knowledge, attitudes, and related practice among male partners in the city of Abha in relation to women’s BC prevention and means for early detection. *Methods*: A cross-sectional study targeting husbands aged 20 years or older chosen from the outpatient clinics in the Urban Primary Health Care Centers in Abha City. Through questionnaires, interview data were collected regarding knowledge about BC and wife practices and attitudes towards BC. *Results*: The study included 832 husbands. The study showed that only 20.2% (95% confidence interval (CI): 19.9–24.1) of husbands had heard about mammography and only 22.1% had heard about breast self-examination among women as a screening test for BC. The most commonly mentioned variations that might occur in relation to BC were size changes (45.6%). The leading source of BC knowledge was from television (48.9%), and the least-mentioned source of information was healthcare workers (22.4%). Husbands mentioned that only 9.3% of wives had been examined before by mammography. In a multivariate logistic regression of male factors associated with wives’ practices and attitudes towards BC, only good knowledge among husbands was a significant factor. *Conclusions*: The study documented the low level of BC knowledge among male partners. There is an urgent need to pay more consideration to disseminating awareness among men, as they are associates, and they must be armed with proper awareness. There is an urgent demand for establishing a national program and educational campaigns towards BC. Stressing the advantages and access to free mammography is necessary.

## 1. Introduction

Breast cancer (BC) is ranked as the most frequently diagnosed cancer site among women in Saudi Arabia [1], with its incidence continuing to increase speedily in recent years [2], and it can occur in women at an early age [3]. Recent Saudi cancer registry data [4] showed that BC was graded first among women, and in 2015, there were 1979 female BC cases. BC amounted to 16.7% of all cancers reported among Saudi nationals, and 30.1% among all cancers reported among women at all ages. The age-standardized incidence rate (ASR) per 100,000 members of the population was 24.3 for the Saudi female population. The incidence appears to differ by geographic regions, with the highest incidence found in the eastern, central, and western regions of the country. The median age at diagnosis was 50 years [4]. A recent study evaluated the burden of BC mortality in Saudi Arabia and mentioned that the deaths attributable to BC are expected to double between 2025 and 2050 [5].

A number of factors have been involved in the etiology of BC [6]. These factors act individually or together to trigger BC. Some BC-associated aspects are preventable, while many are not. Scientifically robust studies have revealed that several lifestyle modifications are likely to prevent BC and can reduce the incidence of this fatal disease and therefore should be suggested [7]. BC has a well-recognized silent stage; documents suggest that the death toll from BC can be substantially decreased by early detection of BC through a screening program. These programs comprise breast self-examination (BSE), clinical breast examination (CBE), and mammograms [8]. Women must be encouraged to gain comprehensive information regarding early detection of BC, learn how to do BSE, and ask for mammography testing [9]. Several studies in Saudi Arabia have reported low awareness of BC and significant obstacles to early presentation among Saudi women [10,11,12]. A key sociocultural barrier against breast screening and early detection of BC in many conservative cultures is that men manage women’s choices and actions. A multiplicity of noneconomic barriers that slow down early detection include cultural, ethnic beliefs, and restrictions. Failure to identify these obstacles can ruin the achievement of any cancer care program, even when sufficient resources are provided [13].

The Aseer region is located in the Southwest of Saudi Arabia. Abha is the capital of Aseer. The aim of this research was to find out the key background knowledge, attitudes, and related practice among male partners in Abha in relation to women’s BC prevention and means for early detection.

## 2. Materials and Methods

### 2.1. Study Design and Target Population

The study design was cross-sectional. The study population was husbands aged 20 years or older. They were chosen from the outpatient clinics in the seven Urban Primary Health Care Centers in Abha. Ethical approval for the study was acquired from the King Khalid University Research Ethics Committee (REC # 20180-03-21) and the Ministry of Health in the Aseer Region.

### 2.2. Sample Size Determination

Utilizing the WHO manual for Sample Size Determination in Health Studies [14] with an anticipated population proportion of 93% [13] and with an absolute precision of 2% at 95% confidence interval, the minimum sample size necessary for the study was computed to be 626 men. To prevent probable loss of cases, a total of 800 persons was originally considered to be included in the study.

### 2.3. Study Tool

The Arabic validated version of the questionnaire used previously in Abha regarding BC knowledge among women was used in the present study [10]. The questionnaire included three areas: sociodemographic background, knowledge about BC, and wife practices and attitudes towards BC. Sociodemographic data contained: age, nationality, occupation, and level of education. Knowledge about BC contained questions about awareness of screening tests for BC (mammography and BSE), changes that might take place in relation to breast cancer (size, heaviness under armpit, nipple discharge, and changes in the shape of the nipple), risk factors that might increase susceptibility (contraceptive pills, hormonal replacement therapy, exposure to excess radiation, smoking, heredity, old age), and protective factors for BC (breast feeding, regular practice of exercise, first pregnancy earlier than 40 years, proper nutrition). Wife practices and attitudes towards BC included history of wife examination before by mammography, history of wife performance of BSE, and wife willingness to be trained more in BSE. The questionnaire also included data regarding sources of knowledge about BC.

### 2.4. Knowledge Scoring System

Each of the 16 items of knowledge about BC was scored 1 if known by the participant or zero if not identified by the participant. The scores were added together.

### 2.5. Data Analysis

Data were coded, validated, and explored using SPSS Software version 22 (IBM Corp, Armonk, NY, USA). For descriptive statistics arithmetic mean, standard deviation (SD) and, 95% confidence intervals (CI) were used to show the precision of the estimate. Binary logistic regression was used to evaluate male partners factors associated with wife practices and attitudes towards BC. Adjusted odds ratios (AOR) and their 95% CI were calculated. Variables included in the model were age, education, nationality, and husband knowledge about BC.

## 3. Results

### 3.1. Description of the Study Sample

The present study included 832 husbands. Their age varied from 20 to 80 years, with an average of 38.2 ± 11.6 (SD) years and a median 35 years. Table 1 shows the sociodemographic profile of the study sample. The majority were Saudis (753, 90.5%) and university-educated (424, 51.0%).

### 3.2. Husbands’ Awareness about Breast Cancer (BC) and Its Accompanying Risk and Protective Factors (Table 2)

The study showed that only 20.2% (95% CI: 19.9–24.1) of husbands had heard about mammography and only 22.1% had heard about breast self-examination among women as a screening test for BC. The most commonly mentioned changes that might occur in relation to BC were size changes (45.6%), changes in the shape of the nipple (45.1%), and discharges from the nipple (41.2%). The most commonly mentioned BC risk factors were heredity (59.9%), smoking (58.1%), and excessive exposure to radiation (52.5%). The most commonly mentioned protective factors for BC were breast feeding (85.2%), proper nutrition (82.1%), and regular practice of exercise (69.2%).

### 3.3. Knowledge Score

The knowledge score ranged from 0 to 16, with an average of 7.8 ± 3.9 (SD) and a median of 8. Those having a score of less than 8 (46.4%) were regarded as having poor knowledge, and the rest were regarded as having good knowledge.

### 3.4. Sources of BC Information

The key source of BC knowledge was from the television (407, 48.9%), the internet (262, 31.5%), and relatives (249, 29.9%). Only 22.4% (186) recognized healthcare workers as their source of knowledge.

### 3.5. Wife Practices and Attitudes Related to Breast Cancer

The study showed that (Table 3), according to their husbands, only 9.3% of wives had been examined before by mammography. Only 17.5% of wives performed BSE. The majority (61.4%) mentioned that wives were willing to be trained more in BSE. In a multivariate logistic regression of male factors associated with wives’ practices and attitudes towards BC (Table 4), only good knowledge among husbands (knowledge score of 8 and more) was a significant factor. For age 40+, university education, and Saudi nationality, all AOR point estimates except one (nationality) were above 1.0 but did not reach statistical significance.

## 4. Discussion

BC in Saudi Arabia is a major public health problem. Early detection of BC will help in minimizing the mortality toll. A recent study in Saudi Arabia showed that BC patients whose tumors were classified as stage IV had the highest mortality rate, which was 5 times higher than that for patients with stage I tumors [15]. Early detection of BC will eventually help in minimizing the burden and death toll of BC. Studies in western countries showed that BC awareness was found to increase the uptake of mammography and BSE behaviors and increase likelihood of BC screening attendance [16]. A recent study in Saudi Arabia identified the factors influencing delayed presentation of BC among Saudi women. The study summarized factors as lack of BC knowledge, wrong beliefs that symptoms would disappear by themselves, and embarrassment and shyness [17].

Recent studies in Saudi Arabia reported low awareness of BC among women in Riyadh [11], Madinah [12], and Hail [18]. Similarly, six years ago, a study in Abha reported the same trend [10]. A systematic review including 56 studies showed the inadequate awareness of women towards BC and pointed out that there were no national breast cancer education programs in Kingdom of Saudi Arabia [19].

To amplify the influences of poor knowledge of women regarding BC as a contributing factor to their lower participation in BC screening activities is the fact that men in Saudi society, to some extent, manage women’s choices and actions. The present study documented the key background knowledge, attitudes, and related practice among male partners in relation to women’s BC prevention and means for early detection. In general, our study pinpointed the low knowledge of male partners regarding BC. Similar findings were reported in Jeddah [13], Jordan [20], and in Ghana [21]. The present study documented the significant role of male partners’ knowledge in wives’ practices and attitudes towards BC.

Regarding BC screening programs (BCSP) in the Kingdom of Saudi Arabia, a recent article revised the previous activities [22]. BCSP started in 2007 via nongovernmental collaboration between the Abdul Lateef Charitable Screening Center and the Saudi Cancer Society and was based in Riyadh. Another nongovernmental BCSP was initiated in the eastern province in Saudi Arabia using mobile mammogram machines between 2009 and 2014. The first governmental breast screening program was a pilot study of a breast screening program in the Al-Qassim region in Saudi Arabia [23]. Nonetheless, a clear description of a nationwide screening program for breast cancer is currently unavailable [22]. A study [24] based on the Saudi Health Interview Survey and investigating utilization of women for breast cancer screening services reported a very low participation rate and concluded that BC screening in Saudi Arabia is free, but there are almost no takers.

Again, the non-utilization rate of available BCSP can be explained by lack of information around BC among women and their male partners, who are the main motivators in the Saudi community in promoting health services utilization if they are well informed and encouraged in this respect.

The present study showed that the media and the TV were the most common sources of BC knowledge among men, and the least common was healthcare workers. Similar results were observed among men in Riyadh [25]. On the other hand, in a study in Jeddah, the main source was healthcare workers, and the least common was the media [13]. In our region, healthcare workers should pay more attention to educating men regarding BC.

Limitations of the present study are related to the small numbers included in the study and the nature of the urban settings, where the near-majority of men were well educated.

## 5. Conclusions

The present study documented the low level of BC knowledge among male partners. Furthermore, husbands’ BC knowledge plays a significant role in wives’ attitudes and practices related to BC. Hence, there is an urgent need to pay more attention to spreading awareness among men, as they are associates, and they need to be armed with proper awareness. We advise also directing BC awareness campaigns towards husbands and men in general. They should be requested to urge their wives and families to join in awareness and screening activities.

There is an urgent demand for establishing a national program and educational campaigns towards BC. Stressing the benefits and access to free mammography is necessary.

## Figures and Tables

**Table 1 ijerph-16-03089-t001:** Sociodemographic characteristic of the study sample of husbands (*n* = 832).

Characteristic	No	%
Age group		
20–39	526	63.2
40–59	249	29.9
60–80	57	6.9
Nationality		
Saudi	753	90.5
Non-Saudi	79	9.5
Educational Level		
Illiterate	41	4.9
Primary–Secondary	367	44.1
University	424	51.0

**Table 2 ijerph-16-03089-t002:** Husbands’ knowledge about breast cancer (BC) and its associated risk and protective factors (*n* = 832).

Knowledge	No (%)	95% CI
Screening tests for BC		
Heard about mammography	168 (20.2%)	19.9–24.1
Heard about BSE	184 (22.1%)	19.4–25.0
Changes that might occur in relation to BC		
Size changes	454 (54.6%)	51.2–57.9
Heaviness under armpit	315 (37.9%)	34.6–41.2
Discharge from the nipple	343 (41.2%)	37.9–44.6
Changes in the shape of the nipple	375 (45.1%)	41.7–48.5
BC Risk Factors		
Use of contraceptive pills	326 (39.2%)	35.9–42.5
Hormonal replacement therapy	302 (36.3%)	33.1–39.6
Exposure to excess radiation	438 (52.6%)	49.2–56.0
Smoking	483 (58.1%)	54.7–61.4
Heredity	498 (59.9%)	56.5–63.1
Old age	336 (40.4%)	37.1–43.7
Protective Factors for BC		
Breast feeding	709 (85.2%)	82.7–87.5
Regular practice of exercise	576 (69.2%)	66.0–72.3
First pregnancy earlier than 40 years	299 (35.9%)	32.7–39.2
Proper nutrition	683 (82.1%)	79.4–84.6

**Table 3 ijerph-16-03089-t003:** Husbands’ knowledge of wife practices and attitudes related to breast cancer.

Wife Practices and Attitudes	No (%)	95% CI
Wife examined before by mammography	77 (9.3%)	7.4–11.4
Wife performed breast self-examination	146 (17.5%)	15.1–22.2
Wife’s willingness to be trained more in BSE	511 (61.4%)	58.1–64.7

**Table 4 ijerph-16-03089-t004:** Adjusted odds ratio (AOR) and 95% confidence intervals (95% CI) of male spouse factors associated with wife practices and attitudes related to breast cancer.

Male Spouse Factors	Wife Examined Before by Mammography AOR (95% CI)	Wife Performed Breast Self-Examination AOR (95% CI)	Wife Willingness to Be Trained More in BSE AOR (95% CI)
Age: 40+ vs. less than 40	1.509 (0.909–2.506)	1.185 (0.997–2.193)	1.146 (0.850–1.543)
Education: University vs. others	1.205 (0.811–2.681)	1.259 (0.988–2.921)	1.133 (0.841–1.526)
Nationality: Saudi vs. non-Saudi	0.976 (0.446–2.187)	1. 124 (0.982–2.785)	1.233 (0.767–1.82)
Knowledge score: good 8+ vs. low less than 8	2.118 (1.231–3.649)	2.330 (1.552–3.498)	1.821 (1.361–2.438)

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
