# Peer review of "Male Partners’ Knowledge, Attitudes, and Perception of Women’s Breast Cancer in Abha, Southwestern Saudi Arabia"

_ijerph, 2019, doi:10.3390/ijerph16173089_

Round 1
Reviewer 1 Report
In this study, Hasan M. Al-Musa and his colleagues investigated male partners’ knowledge, attitudes and perception of women breast cancer in Abha city, southwestern Saudi Arabia. This study is regional but interesting. Though this issue is expectable, the results provide evidence for policy of public health. There are some comments for this study:
There are some grammar errors. Further English editing via native speaker is recommended. Lack of logic writing in the introduction (paragraph 2 to 4). Please integrate these paragraphs and tell a story of your rationale. Lack of logic writing in the discussion (paragraph 2 and 5). Please integrate these paragraphs into other sections. In table 4, the results showed BC knowledge is independent factor for this issue. This is very important information for promoting public education. I recommend the authors emphasized in the conclusion that BC knowledge instead of age, education (school), or nationality plays the major role in this health issue.
Author Response
Responses to reviewer one
The authors are grateful to the constructive comments of the reviewer. Please find point by point response to the comments.
English editing is done. In the introduction section, paragraphs 2 to 4 were integrated (line 49-63). Paragraph 5 in discussion addresses a new idea. It is not easy to be integrated with paragraph 2. In the conclusion section, the following statement was added “Furthermore, husbands BC knowledge plays the significant role in wives’ attitudes and practices related to BC” and necessary modifications were mad. (line174-175).Reviewer 2 Report
The study reports on features highly relevant to the promotion of early diagnosis and prevention of breast cancer among women in the highly paternalistic culture of Saudi Arabia. I have only a few minor recommendations:
line 49: 'extremely' is too strong and may be changed to 'substantially'
line 99: reduce number of digits and specify SD: 38.2 ± 11.6 (SD)
line 110: similarly: 7.8 ± 3.9 (SD)
line115: missing end parenthesis sign: (249, 29.9%)
line 119: 'welling' should be 'willing'
line 121: Suggest adding: 'For age 40+, university education, and Saudi nationality, all aOR point estimates except one were above 1.0, but did not reach statistical significance'.
lines 125, 126: Tumor grade is a histological parameter, and I assume what is referred to is clinical stage and that the wording should be changed accordingly.
line 144: 'Kingdom' should be 'Kingdom of Saudi Arabia'
line 155: 'motivator' should be 'motivators'
line 157: 'was' should be 'were'
line 158 ' least was' should be 'least common was'
line 160: 'least were' should be 'least common was'
Table 1: Education level /illiterate : percentage '49' should be '4.9'
Tables 1 and 2: Oversized digits should be brought in conformity with the rest of the tables
Table 3, table heading: Should be expanded to 'Husbands knowledge of wife practices......)
Table 4: Specify that the figures are aORs. Suggest Table heading is expanded to: 'Adjusted odds ratio and CI of male spouse factors.........'
Author Response
Responses to reviewer two
The authors are grateful to the constructive comments of the reviewer. Please find point by point response to the comments.
The word 'extremely' is changed to 'substantially' (line 54) In line 104: number of digits is revised SD was added: 38.2 ± 11.6 (SD) line 115: number of digits is revised SD was added: 7.8 ± 3.9 (SD) line120: missing end parenthesis sign was added: (249, 29.9%) line 125: 'welling' is changed to 'willing' line 127: 'For age 40+, university education, and Saudi nationality, all aOR point estimates except onewere above 1.0, but did not reach statistical significance' was added. lines 133, 134: The word grade was replaced by stage and that the wording was changed accordingly. line 152: 'Kingdom' was replaced by 'Kingdom of Saudi Arabia' line 163: 'motivator' was replaced by 'motivators' line 165: 'was' was replaced by 'were' line 166 ' least was' was replaced by 'least common was' line 168: 'least were' was replaced by 'least common was' Table 1: Education level /illiterate: percentage '49' was replaced by '4.9' Tables 1 and 2: Oversized digits were brought in conformity with the rest of the tables Table 3, table heading: was rephrased to be “Husbands knowledge of wife practices and attitudes related to breast cancer. Table 4: Heading was rephrased to be “Adjusted odds ratio (aOR) and 95% confidence intervals (95% CI) of male spouse factors associated with wife practices and attitudes related to breast cancer.Round 2
Reviewer 1 Report
The authors answer the questions accordingly.